# The Anti-Cancer Effects of a Zotarolimus and 5-Fluorouracil Combination Treatment on A549 Cell-Derived Tumors in BALB/c Nude Mice

**DOI:** 10.3390/ijms22094562

**Published:** 2021-04-27

**Authors:** Ching-Feng Wu, Ching-Yang Wu, Robin Y.-Y. Chiou, Wei-Cheng Yang, Chuen-Fu Lin, Chao-Min Wang, Po-Hsun Hou, Tzu-Chun Lin, Chan-Yen Kuo, Geng-Ruei Chang

**Affiliations:** 1Division of Thoracic and Cardiovascular Surgery, Department of Surgery, Chang Gung Memorial Hospital, Chang Gung University, Linkou, 5 Fuxing Street, Guishan District, Taoyuan 33305, Taiwan; maple.bt88@gmail.com (C.-F.W.); wu.chingyang@gmail.com (C.-Y.W.); 2Department of Food Science, National Chiayi University, 300 University Road, Chiayi 60004, Taiwan; rychiou@mail.ncyu.edu.tw; 3Department of Veterinary Medicine, School of Veterinary Medicine, National Taiwan University, 4 Section, 1 Roosevelt Road, Taipei 10617, Taiwan; yangweicheng@ntu.edu.tw; 4Department of Veterinary Medicine, College of Veterinary Medicine, National Pingtung University of Science and Technology, 1 Shuefu Road, Neipu, Pingtung 912301, Taiwan; cflin2283@mail.npust.edu.tw; 5Department of Veterinary Medicine, National Chiayi University, 580 Xinmin Road, Chiayi 60054, Taiwan; leowang@mail.ncyu.edu.tw (C.-M.W.); lin890090@gmail.com (T.-C.L.); 6Department of Psychiatry, Taichung Veterans General Hospital, 4 Section, 1650 Taiwan Boulevard, Taichung 40705, Taiwan; peterhopo2@yahoo.com.tw; 7Faculty of Medicine, National Yang-Ming University, 2 Section, 155 Linong Street, Beitou District, Taipei 11221, Taiwan; 8Department of Research, Taipei Tzu Chi Hospital, Buddhist Tzu Chi Medical Foundation, 289 Jianguo Road, Xindian District, New Taipei City 231405, Taiwan

**Keywords:** apoptosis, 5-Fluorouracil, lung adenocarcinoma, inflammation, metastasis, zotarolimus

## Abstract

Zotarolimus is a semi-synthetic derivative of rapamycin and a novel immunosuppressive agent used to prevent graft rejection. The pharmacological pathway of zotarolimus restricts the kinase activity of the mammalian target of rapamycin (mTOR), which potentially leads to reductions in cell division, cell growth, cell proliferation, and inflammation. These pathways have a critical influence on tumorigenesis. This study aims to examine the anti-tumor effect of zotarolimus or zotarolimus combined with 5-fluorouracil (5-FU) on A549 human lung adenocarcinoma cell line implanted in BALB/c nude mice by estimating tumor growth, apoptosis expression, inflammation, and metastasis. We established A549 xenografts in nude mice, following which we randomly divided the mice into four groups: control, 5-FU (100 mg/kg/week), zotarolimus (2 mg/kg/day), and zotarolimus combined with 5-FU. Compared the results with those for control mice, we found that mice treated with zotarolimus or zotarolimus combined with 5-FU retarded tumor growth; increased tumor apoptosis through the enhanced expression of cleaved caspase 3 and extracellular signal-regulated kinase (ERK) phosphorylation; decreased inflammation cytokines levels (e.g., IL-1β, TNF-α, and IL-6); reduced inflammation-related factors such as cyclooxygenase-2 (COX-2) protein and nuclear factor-κB (NF-κB) mRNA; enhanced anti-inflammation-related factors including IL-10 and inhibitor of NF-κB kinase α (IκBα) mRNA; and inhibited metastasis-related factors such as transforming growth factor β (TGF-β), CD44, epidermal growth factor receptor (EGFR), and vascular endothelial growth factor (VEGF). Notably, mice treated with zotarolimus combined with 5-FU had significantly retarded tumor growth, reduced tumor size, and increased tumor inhibition compared with the groups of mice treated with 5-FU or zotarolimus alone. The in vivo study confirmed that zotarolimus or zotarolimus combined with 5-FU could retard lung adenocarcinoma growth and inhibit tumorigenesis. Zotarolimus and 5-FU were found to have an obvious synergistic tumor-inhibiting effect on lung adenocarcinoma. Therefore, both zotarolimus alone and zotarolimus combined with 5-FU may be potential anti-tumor agents for treatment of human lung adenocarcinoma.

## 1. Introduction

The vast majority of lung cancers are caused by epithelial cell lesions resulting from malignant epithelial tumors of the lung. Lung cancer is a leading cause of cancer-related deaths among men and women worldwide [1]. Globally, lung cancer causes about 1.8 million deaths each year [2]. Common symptoms include shortness of breath, coughing (hemoptysis), and weight loss. Lung cancer is mainly divided into two broad histological subtypes: non-small-cell lung cancer (NSCLC), including lung adenocarcinoma and lung squamous-cell carcinoma, and small-cell lung cancer (SCLC). These are categorized by the type of cell in which the cancer originates [3]. The most common type of lung cancer is NSCLC, which has high morbidity and mortality rates [4]. Related treatments and prognosis plans are created on the basis of the histological type of cancer, cancer stage, and patient status. Possible treatment options are surgery, chemotherapy, and radiotherapy [3,5]. Chemotherapy is the application of chemicals or drugs to kill cancer cells. Commonly administered anticancer drugs include alkylating agents (e.g., cisplatin), mitotic inhibitors (e.g., paclitaxel), epidermal growth factor receptor (EGFR) inhibitors (e.g., gefitinib), vascular endothelial growth factor (VEGF), or VEGF receptor inhibitors (e.g., bevacizumab) [6].

The mammalian target of rapamycin (mTOR) is a serine-threonine protein kinase expressed in all cells. mTOR regulates cell transcription, translation, cell growth, cell arrangement, cell differentiation, and apoptosis [7]. The mTOR pathway is important for growth development, and its signaling is involved in insulin signaling, growth factors, nutrition, energy metabolism, obesity, cancer, and other diseases [8,9,10]. Tuberous sclerosis 1 and 2 (TSC1 and TSC2) are upstream of the mTOR pathway and can hydrolyze GTP into GDP, rendering mTOR inactive and blocking the mTOR signaling pathway. However, activated mTOR phosphorylates S6K (ribosomal protein S6 kinase, p70S6K), which initiates a series of related cellular physiological responses including vascular and tumor cell proliferation and other pathways related to tumor formation [11]. mTOR activation promotes protein translation, enhances cell growth, and affects intracellular metabolism [12]. Therefore, the development of new drugs to inhibit mTOR activation is important, and these novel drugs can serve as new approaches for the treatment of tumors.

Zotarolimus is an analogue of rapamycin, an immunosuppressive drug. Rapamycin prevents allograft rejection and is used in drug-eluting stents (DESs) to reduce post-angioplasty restenosis [13,14]. Rapamycin, however, can produce adverse effects such as immunodeficiency and hypertension as well as cardiac and vascular, lipid metabolic, testicular and epididymal, dermatological, obstetric and gynecological, ocular, and neurological problems [15,16]. Continual administration of zotarolimus is safer than that of rapamycin, with lower risks for metabolic, diabetic, and hyperclycemic adverse events [9,16,17]. Zotarolimus was the first cytostatic agent to be used primarily for DES delivery [18]; in zotarolimus, a tetrazole ring replaces the hydroxyl group found at position 42 in rapamycin, which makes zotarolimus highly lipophilic. As a result, zotarolimus has a significantly greater octanol-water partition coefficient than other drugs used in DESs. Drug-loaded stents enable drugs from the stent to be gradually released into the coronary artery wall and the properties of zotarolimus provide several advantages on this application. Additionally, its lipophilic nature may favor the crossing of cell membranes to restrict the neointimal proliferation of target tissues. A majority of research on the use of zotarolimus focuses on the treatment of cardiovascular diseases, and little research has been addressed to its potential use for other diseases [19,20]. Everolimus, also a mTOR inhibitor, reduces the expression of VEGF in tumor-derived mouse ovarian carcinoma and in gastric cancer cells in vitro. In vivo, everolimus considerably restricts tumor growth [21,22]. In addition, everolimus has been used to treat pancreatic, bladder, and lung cancers [23,24,25]. Zotarolimus, exhibiting a similar mechanism of action to everolimus, could be used in the treatment of lung cancer [26]. For lung cancer treatment, 5-fluorouracil (5-FU) is also widely used [27], although potential drug resistance limits the clinical use of 5-FU. Studies have reported that the anticancer efficacy of 5-FU increased with the dose [28,29], but the cytotoxic effect on normal cells induced unacceptable levels of toxicity in patients [30]. To overcome this, 5-FU should be combined with other anticancer drugs with different mechanistic actions [28].

Patients with lung cancer are mainly treated with chemotherapy. However, certain patients do not respond to this therapy or may respond well initially but gradually show signs of a relapse. This results in increased drug doses, which may cause adverse drug reactions or the development of drug resistance [31]. Thus, developing new drugs or a combination of drugs for the treatment of lung cancer is crucial. Although the pyrimidine analog 5-FU is widely used to treat cancers, consequent drug resistance seriously limits its clinical use in lung cancer treatment [27]. Therefore, this study examined the antitumor effects of the mTOR inhibitor zotarolimus on A549 cells as well as the synergistic effects of zotarolimus when combined with 5-FU. The examination was conducted on an A549 human lung adenocarcinoma cell line implanted into BALB/c nude mice. We observed tumor growth and various aspects of factors related to tumor development, including apoptosis, inflammation, and metastasis, to explore the therapeutic properties of zotarolimus and mTOR drugs for lung cancer and thereby increase the number of available drugs for lung cancer treatment choices.

## 2. Results

### 2.1. Zotarolimus Retards Tumor Growth

The previous study suggests that 5-FU and zotarolimus have anticancer activity [27,28]. Accordingly, we observed reduced tumor growth and volume in the groups treated with 5-FU alone, zotarolimus alone, and a combination of zotarolimus and 5-FU compared with the control group (Figure 1a). In addition, the time course of responses revealed a significant greater decrease in tumor growth in the mice treated with 5-FU (*p* < 0.001), zotarolimus (*p* < 0.05), and zotarolimus combined with 5-FU (*p* < 0.001) than that in the control mice (Figure 1b). Combining zotarolimus with 5-FU led to a greater decrease in tumor growth, representing an obvious synergistic function. Compared to the control group, the tumor inhibition rate increased by 50.0% in the 5-FU-treated group, 31.2% in the zotarolimus-treated group and 66.7% in the group treated with zotarolimus combined with 5-FU (Table 1). Therefore, zotarolimus combined with 5-FU was a powerful inhibitor of the growth of lung adenocarcinoma.

### 2.2. Zotarolimus Increases the Number of TUNEL-Positive Cells

The TUNEL assay is designed to detect apoptotic cells, which are subjected to extensive DNA degradation during the late apoptosis stages [7]. Our findings indicated that TUNEL positive cells increased 5.5 (*p* < 0.001), 3.3 (*p* < 0.001), and 9.2 (*p* < 0.001) times (Figure 2b) in mice treated with 5-FU, zotarolimus, and zotarolimus combined with 5-FU, respectively (Figure 2a). These values were significantly greater than those observed for the control group. The zotarolimus-treated group exhibited significantly lower TUNEL-positive cells than did the 5-FU-treated group. Moreover, the number of TUNEL-positive cells in mice treated with zotarolimus-only group was lower than those in mice treated with 5-FU-only group (by 40.8%) and the zotarlimus and 5-FU combined group (by 64.3%). These results indicated that both zotarolimus and a combination of zotarolimus and 5-FU increased the degree of apoptosis, and the combination treatment with zotarolimus and 5-FU had the greatest effect.

### 2.3. Zotarolimus Increases Apoptosis-Related Protein Expression

Apoptotic activation results in the activation or inactivation of different substrates, stimulating a cascade of signaling events that control the degradation of cellular components [32]. We selected apoptosis-related proteins, including cleaved caspase 3, extracellular signal-regulated kinase (ERK), phosphorylated ERK, and anti-apoptosis protein B-cell lymphoma 2 (Bcl-2), for analysis (Figure 3a). Using western blotting, we found that the expression of cleaved caspase 3 and phosphorylated ERK was significantly higher in the groups treated with 5-FU (cleaved caspase 3: *p* < 0.001; phosphorylated ERK: *p* < 0.001) and zotarolimus combined with 5-FU (cleaved caspase 3: *p* < 0.001; phosphorylated ERK: *p* < 0.001) than in the control group (Figure 3b,c). In addition, zotarolimus-treated mice had a higher level of cleaved caspase 3 (*p* < 0.001) and ERK phosphorylation (*p* < 0.01) than control mice. In contrast, the 5-FU and zotarolimus-treated groups had a lower expression of cleaved caspase 3 (5-FU: *p* < 0.01; zotarolimus: *p* < 0.001) and ERK phosphorylation (5-FU: *p* < 0.001; zotarolimus: *p* < 0.001) than zotarolimus combined with 5-FU-treated group. Mice treated with zotarolimus combined with 5-FU had the highest expression of cleaved caspase 3 and ERK phosphorylation. Bcl-2 is widely considered a suppressor of apoptosis. Bcl-2 expression in the 5-FU (*p* < 0.001) and zotarolimus groups (*p* < 0.001) was lower than in the control group (Figure 3d). However, Bcl-2 expression in the zotarolimus group was higher than in the 5-FU-treated group (*p* < 0.01) and in the group treated with zotarolimus combined with 5-FU (*p* < 0.001), which had the lowest Bcl-2 expression. The results indicated that zotarolimus and zotarolimus combined with 5-FU could enhance the expression of apoptosis-related proteins and inhibit anti-apoptotic protein expression. Furthermore, zotarolimus combined with 5-FU exhibited the maximum effect on apoptosis.

### 2.4. Zotarolimus Inhibits Production of Cytokines

Resisting cancer-linked inflammation can delay the disease’s progression. The histopathology, observed using immunohistochemistry, revealed levels of inflammatory cytokines including IL-1β (Figure 4a) and TNF-α (Figure 4b). After quantification, in comparison with the control group, the expression of IL-1β and TNF-α were lower in mice treated with 5-FU (by 46.0% and 44.7%), zotarolimus (27.8% and 18.5%), and zotarolimus combined with 5-FU (89.4% and 81.1%) (Figure 4d). However, the activity of IL-1β and TNF-α was 1.3 (*p* < 0.01) and 1.5 (*p* < 0.001) times higher in the zotarolimus group than in the 5-FU group. Moreover, IL-1β and TNF-α expression were the lowest in mice treated with zotarolimus combined with 5-FU. This combination treatment resulted in lower expression than in mice treated with zotarolimus alone (by 85.3% for IL-1β and 76.8% for TNF-α) or 5-FU alone (by 80.3% for IL-1β and 65.8% for TNF-α). Thus, the expression of IL-1β and TNF-α in mice treated with zotarolimus combined with 5-FU could be largely suppressed in lung adenocarcinoma.

In addition, we examined the levels of the inflammatory cytokine IL-6 and the anti-inflammatory cytokine IL-10 in serum using an enzyme-linked immunosorbent assay (ELISA). The expression trend for IL-6 was similar to those of IL-1β and TNF-α. The levels of IL-6 in groups treated with 5-FU, zotarolimus, and zotarolimus combined with 5-FU, respectively, were 80.6%, 50.6%, and 94.2% lower than that of the control group (Figure 4e). However, the IL-6 levels in mice treated with zotarolimus were 2.6 times higher than those in mice treated with 5-FU. The level of IL-10 in zotarolimus-treated mice was 19.0% lower than that of IL-10 in 5-FU-treated mice (Figure 4f). Notedly, the decrease in the expression of IL-6 due to zotarolimus combined with 5-FU was 69.6% and 88.1% greater than the decline shown by 5-FU or zotarolimus treatments alone. The level of IL-10 in mice treated with zotarolimus combined with 5-FU was the highest (by 108.7% for the 5-FU group and 134.1% for the zotarolimus group). Thus, a reduced expression of IL-1β, TNF-α, and IL-6 and an enhanced expression of IL-10 were observed in lung adenocarcinoma in mice treated with zotarolimus combined with 5-FU.

### 2.5. Zotarolimus Inhibits Inflammation-Related Factors

Inflammation is closely linked to cancer, and its reduction or elimination may lead to more effective cancer prevention and therapy strategies [33]. Using a western blot, we determined that the cyclooxygenase-2 (COX-2) expression was lower in the groups treated with 5-FU (*p* < 0.001), zotarolimus (*p* < 0.001), and zotarolimus combined with 5-FU (*p* < 0.001) than in the control group (Figure 5a). Zotarolimus- and 5-FU-treated mice reported a significantly reduced expression of COX-2 compared to control mice. However, the expression of COX-2 was higher in zotarolimus-treated mice (*p* < 0.001) than that in the 5-FU-treated mice (Figure 5b). The treatment with zotarolimus combined with 5-FU resulted in the lowest COX-2 expression. The stimulation of cells with inflammatory cytokines results in the degradation of IκB kinase (IκBα), which inhibits nuclear factor κB (NF-κB). This leads to the accumulation of NF-κB and regulation of the specific gene expression [34]. We detected the expression of NF-κB and IκBα mRNA and indicated that NF-κB was significantly lower in groups treated with 5-FU (*p* < 0.001), zotarolimus (*p* < 0.001), and zotarolimus combined with 5-FU (*p* < 0.001) than in the control group. Further, the inhibitory effect on NF-κB expression in zotarolimus-treated mice (*p* < 0.01) was higher than that in 5-FU-treated mice (Figure 5c). Zotarolimus combined with 5-FU largely reduced the expression of NF-κB in all groups. As expected, mice treated with 5-FU (*p* < 0.001), zotarolimus (*p* < 0.001), and zotarolimus combined with 5-FU (*p* < 0.001) reported a higher expression of IκBα than the control mice (Figure 5d). However, the expression of IκBα was lower (*p* < 0.05) in the zotarolimus group than in the 5-FU group. Zotarolimus combined with 5-FU increased the expression of IκBα mRNA. Our results showed that zotarolimus inhibited inflammation-related factors and inflammatory responses. However, zotarolimus-treated mice did not show a greater reduction in inflammation-related factors than 5-FU-treated mice. Nevertheless, the co-effect of zotarolimus and 5-FU achieved a greater inhibition of the inflammatory reactions in mice.

### 2.6. Zotarolimus Inhibits Metastasis-Related Factors

Cancer metastasis is the major cause of cancer morbidity and mortality and accounts for approximately 90% of cancer deaths [35]. For observations, we selected metastasis-related factors including TGF-β, CD44, EGFR, and VEGF. We used IHC to detect the expression of TGF-β and CD44 (Figure 6a,b). The pathological sections indicated lower TGF-β and CD44 expression in the groups treated with 5-FU (TGF-β: *p* < 0.001; CD44: *p* < 0.001), zotarolimus (TGF-β: *p* < 0.001; CD44: *p* < 0.001), and zotarolimus combined with 5-FU (TGF-β: *p* < 0.001; CD44: *p* < 0.001) than in the control group. Quantification of the results showed that, compared with the control group, the levels of TGF-β expression in those treated with 5-FU, zotarolimus, and zotarolimus combined with 5-FU were lower by 63.1%, 29.2%, and 81.0%, respectively (Figure 6c). Further, the levels of CD44 expression in the groups treated with 5-FU, zotarolimus, and zotarolimus combined with 5-FU were lower by 64.1%, 31.1%, and 73.9%, respectively (Figure 6d). However, the TGF-β and CD44 expression in the zotarolimus group were both 1.9 times higher than those in the 5-FU group. The group treated with zotarolimus combined with 5-FU showed significantly reduced TGF-β and CD44 expression in comparison with those reported for the 5-FU group and zotarolimus group (by 48.4% and 73.1% for TGF-β and by 27.4% and 61.6% for CD44). The western blotting revealed similar trends for VEGF and EGFR (Figure 6e). The expression of EGFR and VEGF showed a significantly greater decrease in the groups treated with 5-FU (EGFR: *p* < 0.001; VEGF: *p* < 0.01), zotarolimus (EGFR: *p* < 0.01; VEGF: *p* < 0.01), and zotarolimus combined with 5-FU (EGFR: *p* < 0.001; VEGF: *p* < 0.001) when compared with the control group (Figure 6f,g). The EGFR expression in zotarolimus-treated mice (*p* < 0.01) was higher than that in 5-FU-treated mice; however, the VEGF expression in the zotarolimus group was not significant in comparison with the 5-FU group. Mice treated with zotarolimus combined with 5-FU showed the lowest expression of EGFR and VEGF. Thus, zotarolimus could inhibit metastasis-related factors, and zotarolimus combined with 5-FU had the greatest suppressing effect on metastasis.

## 3. Discussion

This study examined the impact of zotarolimus alone and zotarolimus combined with 5-FU on the development of A549 tumors in nude mice. Nude mice constitute an effective model for evaluating the ectopic xenotransplantation of human carcinoma cells for drug or chemical treatment [7,36,37]. Tumor cell–bearing nude mice could be favorable for tumor growth because such mice have no thymus to produce T cells [38]. A preliminary experiment showed that zotarolimus (1 mg/kg/day) significantly inhibit the growth of a lung adenocarcinoma tumor formed from A549 cells, and zotarolimus combined with 5-FU provided a similar inhibitory effect on tumor weight as 5-FU alone (Figure A1). Therefore, we decided to use 2 mg/kg/day of zotarolimus. The results indicated zotarolimus inhibited tumor size and weight, as was observed for the 5-FU-treated mice. However, the inhibitory effect of zotarolimus alone was not greater than that of 5-FU alone. Zotarolimus combined with 5-FU achieved the highest inhibitory effect when compared with groups treated with 5-FU or zotarolimus alone. In terms of apoptosis, zotarolimus combined with 5-FU increased the expression of cleaved caspase 3 and ERK phosphorylation and decreased Bcl-2 expression. This indicated that zotarolimus combined with 5-FU could be the most effective in promoting apoptosis during lung adenocarcinoma treatment. From the viewpoint of inflammation, both groups treated with zotarolimus and zotarolimus combined with 5-FU reduced the production of inflammatory factors including IL-1β, TNF-α, IL-6, COX-2, and NF-κB. Finally, zotarolimus inhibited metastasis-related factors including TGF-β, CD44, VEGF, and EGFR. Accordingly, the synergistic effect of zotarolimus and 5-FU largely inhibited the expression of inflammatory and tumor metastasis-related factors.

The mTOR pathway is a major downstream signaling pathway activated through multiple biological functions involved in cell cycle regulation and performs key functions in autophagic regulation [39,40]. Signals from aberrantly activated mTOR not only promote the growth and metastasis of tumor cells but also help the cells invade healthy tissues [41]. Many drugs inhibiting the mTOR pathway are used in cancer therapy, including temsirolimus, everolimus, and ridaforolimus [42,43]. A study on the effectiveness of temsirolimus demonstrated that the drug inhibits the proliferation of cultured NSCLC cells at a low concentration and prolongs the survival of mice with disseminated pleural NSCLC tumors [44]. Everolimus treatment markedly delayed tumor development and inhibited the expression of VEGF in tumor-derived cell lines from ovarian cancers. In addition, everolimus effectively inhibits the phosphorylation of p70S6K and upregulates the phosphorylation of protein-serine-threonine kinase (AKT) [45]. A study also showed that ridaforolimus inhibits mTOR activity, and thus, tumor cell proliferation and VEGF production [46]. In sum, these studies have highlighted that mTOR pathway inhibitors are effective in cancer treatment, which has furthered research interest in their nature and application. Our results showed that zotarolimus inhibited the growth of A549 cells, and thus, it could be suggested in clinical administered to inhibit the proliferation of lung adenocarcinoma.

Several studies have shown that combination chemotherapy with an anti-cancer drug and a cell-signal inhibitor achieves a better response rate than either used alone. Everolimus, for example, in combination with a high dose of cyclophosphamide showed synergistic antitumor activity in vivo, as observed in the treatment of gastric cancer [47]. Furthermore, addition of rapamycin diminished the residual activity of mTORC1 and significantly intensified the cytotoxicity of cisplatin in the gastric carcinoma cells producing alpha-fetoprotein (AFP) to be resistant to cisplatin [48]. Therefore, this study investigated the inhibitory and synergistic impact of zotarolimus combined with 5-FU on A549 lung adenocarcinoma cells implanted in mice. Our results revealed that zotarolimus combined with 5-FU had a strong inhibitory effect on tumor cell growth. In addition, zotarolimus alone had inhibitory effects on A549 cells. The synergistic effect of zotarolimus and 5-FU further strengthened the inhibitory potency on tumor growth.

Next, we explored the apoptotic fraction, an important indicator for tumors. Cancer cells can evade apoptosis and continually replicate despite abnormalities. Chemotherapy drugs and radiation force the apoptosis of cancer cells, wherein death signals are triggering by DNA damage or cellular distress [49]. The findings related to TUNEL and DAPI for mice treated with zotarolimus and zotarolimus combined with 5-FU revealed the extent of apoptosis in A549 lung adenocarcinoma cells. Zotarolimus enhanced the apoptosis of lung adenocarcinoma, even without 5-FU. Zotarolimus combined with 5-FU substantially increased the fraction of apoptotic cells. Thus, zotarolimus combined with 5-FU hinders the ability of cancer cells to evade apoptosis, increasing the rate of apoptosis and inhibiting tumor growth. In addition, our Western blots showed higher expression of cleaved caspase 3 and ERK phosphorylation, which are central players in apoptosis, in mice treated with zotarolimus combined with 5-FU. Bcl-2 expression exhibited an opposing trend because tumor cells become dependent on anti-apoptotic Bcl-2 to survive [50]. One study demonstrated that treatment with mTOR kinase inhibitors resulted in elevated expression levels of cleaved caspase 3 proteins and dramatically downregulated Bcl-2 levels when mTOR was downregulated in human laryngocarcinoma and lung cancer [51,52]. Everolimus also significantly decreased the expression of all pro-survival Bcl-2 family proteins, including p-Bcl-2 (Ser70), p-Bcl-2 (Thr56), and Bcl-2 [53]. Moreover, zotarolimus also reduces the expression of S6K1 and mTOR (Figure A2). One downstream pathway of mTOR targets S6K, a protein which attaches to mitochondrial membranes and can phosphorylate serine 136 in the pro-apoptotic molecule BAD (Bcl-xL/Bcl-2 associated death promoter) to inactivate it. This phosphorylation interrupts BAD’s binding to the mitochondrial death inhibitors Bcl-XL and Bcl-2 [54]. Thus, mTOR can inhibit apoptosis. Accordingly, through the inhibition of the mTOR pathway, the expression of cleaved caspase 3 and ERK phosphorylation showed a significant higher increase in the groups treated with zotarolimus and zotarolimus combined with 5-FU than in the control group. The apoptosis-related results showed a corresponding trend for tumor size in line with our expectations.

There is significant research and clinical data that support the interdependent relationship between inflammation and tumors [55]. Chronic inflammation has been closely associated with tumorigenesis, and several solid tumors maintain an inflammatory immune microenvironment to accelerate tumor progression and metastasis [56]. We observed inflammation factors such as IL-1β, TNF-α, IL-6, IL-10, COX-2, and NF-κB. IL-1β and TNF-α, in particular, show a notable overlap of biological activity in cancer and promote tumor angiogenesis by inducing tumor cells to secrete VEGF via the IκB/TSC1/mTOR pathway [57]. Studies on lung cancer patients have shown a relationship between IL-6 and cancer cachexia and cancer-related fatigue [58]. IL-6 is a pro-tumorigenic cytokine that triggers JAK/STAT3 activation, which promotes tumor cell growth and suppresses tumor cell apoptosis [59]. The anti-inflammatory response is largely controlled by IL-10. Related to the effect of anti-inflammatory responses, the binding of IL-10 to IL-10R activates the IL-10/JAK/STAT3 cascade, where phosphorylated STAT3 homodimers translocate to the nucleus within seconds, indirectly influencing the expression of target genes such as TNF-α and NF-κB [60]. IL-10 inhibits tumorigenesis by downregulating VEGF, IL-1b, TNF-α, IL-6, and MMP-9 [61]. Research has associated IL-10 deficiency with the increased production of pro-inflammatory cytokines, which promote tumor growth in mice [62]. Here, the administration of zotarolimus could reduce the expression of IL-1β, TNF-α, and IL-6 and increase serum IL-10 levels. In addition, zotarolimus combined with 5-FU significantly reduced inflammation-related factors and increased anti-inflammation factors. C-reactive protein (CRP) synthesis mainly occurs in response to stimulation by pro-inflammatory cytokines [63]. Elevated levels of CRP have been associated with increased lung cancer risk and tumor progression [64]. We found that, in comparison with the control mice, mice treated with 5-FU, zotarolimus, or zotarolimus combined with 5-FU reported reduced serum CRP (Figure A3). The highest level of serum CRP was exhibited in the group administered with zotarolimus combined with 5-FU, which was associated with the highest inhibition rate of tumor growth. Thus, zotarolimus can retard tumor growth by attenuating inflammatory cytokine expression and elevating anti-inflammation mediators.

The TNF-α stimulation activates IκB, a major downstream kinase in the TNF-α signaling pathway, where it phosphorylates IκBα and, subsequently, triggers its degradation through the ubiquitin-proteasome proteolytic system [65]. In the absence of IκBα, which is necessary to retain NF-κB in the cytosol, NF-κB is liberated and translocates into the nucleus and activates other target genes, such as genes encoding cytokines and those involved in angiogenesis, proliferation, and metastasis [66]. Various NF-κB target genes inhibit apoptosis and mediate inflammation, and thus, NF-κB functions as a tumor promoter in inflammation-associated cancer [67]. In malignant cells, NF-κB inhibition would increase susceptibility to apoptosis-inducing agents [66]. Accordingly, NF-κB and IκBα play a role in various immune processes and inflammatory responses. We also detected their mRNA expression in A549 lung adenocarcinoma. Zotarolimus and zotarolimus combined with 5-FU increased the expression of IκBα and inhibited NF-κB, and subsequently, prevented inflammatory reactions. This result showed a reduction in the NF-κB transcription factor family, which inhibited the central mediator of the inflammatory process and enhances a key participant in the immune response to cancer [68]. Thus, zotarolimus could affect the development of tumors by regulating the critical link between inflammation and cancer. Proliferation of cell nuclear antigen (PCNA) is also an indicator of cell proliferation activity, and its expression is correlated with the development of cancer and positively correlated with COX-2 [69]. Studies have shown that the elevated levels of COX-2 have implications in angiogenesis, tumor invasion, resistance to apoptosis, and suppression of antitumor immunity [70]. Similar to PCNA, the expression of Ki67 is strongly associated with tumor cell proliferation and growth and is widely used in routine pathological investigation as a proliferation marker [71]. Our study indicated that zotarolimus alone and zotarolimus combined with 5-FU significantly decreased the expression of PCNA (Figure A4) and Ki67 (Figure A5) in lung adenocarcinoma. Thus, zotarolimus inhibits the expression of growth and anti-apoptotic factors in lung adenocarcinoma cell by downregulating inflammation and tumorigenesis during the stages of tumor promotion and progression.

Another key indicator in oncology research is metastasis. Studies have shown that TGF-β1 stimulates the proliferation and migration of highly transformed tumor cells, causing metastasis and tumor progression [72]. In addition, TGF-β decreases the cytotoxicity of anticancer drugs [73]. CD44 regulates the phosphoinositide-3-kinase (PI3K)/AKT/mTOR signaling pathway and promotes cancer cell migration [74]. CD44-induced epithelial-mesenchymal transition (EMT), however, is strongly correlated with cancer metastasis and is regulated by TGF-β1, and the anti-tumor effect could inhibit the expression of both CD44 and TGF-β [75]. Our results showed that zotarolimus and zotarolimus combined with 5-FU largely reduced the presence of TGF-β and CD44 in tumor immunostaining. Therefore, zotarolimus and zotarolimus combined with 5-FU reduced the metastasis ability of the A549 lung adenocarcinoma cells. Accordingly, it is likely that reducing the expression of TGF-β rendered the effect of drug therapy more prominent, causing inhibition on tumor metastasis, which is in line with the abovementioned reference [73]. As for other metastasis-related factors, VEGF is a potent promoter of angiogenesis, and the overexpression of VEGF is associated with the progression and metastasis of tumor progression [76]. A study on the effect of mTOR on VEGF showed that everolimus reduced human VEGF secretion by 20–45% in wild type and resistant cancer cell lines including GEO and GEO-GR (gefitinib resistant) colon cancer [77]. A similar report indicated that the inhibition of tumors by rapamycin is based on antiangiogenic activity, which is correlated with the impaired production of VEGF and the blockage of vascular endothelial cell stimulation induced by VEGF [78]. EGFR is a major signal transducer of mitogens in cancer pathogenesis and the progression, upstream, of mTOR and is an important target in anticancer therapy [79]. VEGF and EGFR signaling synergize to promote epidermal tumor growth [80]. In addition, zinc finger E-box-binding homeobox 1 (ZEB1) acts as an oncogene in invasive and metastatic lung cancer cells, in which the ZEB1-induced epithelial–mesenchymal transition promotes the loss of epithelial cell polarity and adhesion; induces cytoskeleton remodeling; and drives growth, migration, invasion, and metastasis [81,82]. Zotarolimus alone and zotarolimus combined with 5-FU reduced the metastatic ability of the A549 lung adenocarcinoma cells, which may be attributed to the decreased mRNA expression of ZEB1 (Figure A6). Our results indicate zotarolimus has an anti-tumor effect, and when zotarolimus is combined with 5-FU, this effect on suppressing tumor metastasis is strengthened. Thus, zotarolimus combined with 5-FU significantly suppressed tumors by reducing VEGF and EGFR expression.

An increasing number of lung cancers are becoming resistant to chemotherapy drugs and forming resistant cell lines. Cisplatin, for example, is a common constituent of first-line treatment after surgery [83,84]. However, many patients undergoing lung cancer treatment have shown resistance to cisplatin on the basis of several alterations or defects in signaling pathways in response to the DNA damage caused by cisplatin [85]. Recently, a combination of BEZ235, a novel dual PI3K/mTOR inhibitor, and cisplatin was reported to provide synergistic antitumor effects in cisplatin-resistant A549 cells, as reflected by reduced proliferation, increased apoptosis, and migration suppression. Moreover, tumor development and metastasis are closely related to the structure and function of the tumor microenvironment (TME) [86]. Tumor-infiltrating lymphocytes (TILs) are reportedly involved in lung cancer development, prognosis, and immunotherapy efficacy in the TME [87]. During the IHC staining of tumor tissues with CD45 (TIL-specific marker), the percentages of CD45 activity in the 5-FU, zotarolimus, and zotarolimus combined with 5-FU groups were 9.5-fold, 8.6-fold, and 10-fold higher than that in the control group, respectively (Figure A7) [88]. These data clearly indicate that zotarolimus plays a key role in the inhibition of lung adenocarcinoma. In addition, the mechanism mediating these effects may be associated with the inhibition of PI3K/Akt/mTOR signaling [89]. The anti-tumor effects of zotarolimus alone on tumor growth were not superior to those of 5-FU in the A549 human lung adenocarcinoma epithelial cells. However, zotarolimus combined with 5-FU exhibited excellent inhibition of A549 cell growth in terms of apoptosis, inflammation, and metastasis.

## 4. Materials and Methods

### 4.1. Animals and Cell Lines

For the purpose of this research, six-week-old male BALB/cByJNarl mice were purchased from the National Laboratory Animal Breeding and Research Center (Taipei, Taiwan). Each cage housed two mice, and sterilized food and water were provided. The mice were kept at a constant temperature (22 ± 2 °C) and relative humidity (55 ± 5%) under a 12:12 h light:dark cycle. The animal use protocol was reviewed and approved by the Institutional Animal Care and Use Committee (IACUC) at the National Chiayi University (IACUC Approval No. 108028). All procedures were in line with the Guidelines for the Care and Use of Laboratory Animals by Taiwan’s Ministry of Health and Welfare.

The A549 human lung carcinoma cell line was purchased from the Bioresource Collection and Research Center (Hsinchu, Taiwan). The cells were grown in Dulbecco’s modified Eagle’s medium and supplemented with 5% fetal bovine serum, 50 IU/mL of penicillin, and 50 mg/mL of streptomycin (Gibco Laboratories, Grand Island, NY, USA) in a humidified atmosphere of 95% air and 5% CO_2_ at 37 °C. The cells were routinely passaged by removing the medium and overlaying the cell monolayer with 0.25% trypsin and 0.1% EDTA.

### 4.2. Tumor Inoculation and Treatment

The experiment was designed such that the treatment began after the injection of A549 cells formed a tumor mass detectable on day 7. Briefly, once the mice were anesthetized using an intraperitoneal injection of Zoletil^®^ (Virbac Taiwan, Taipei, Taiwan), a subcutaneously injection in the posterior leg delivered 100 µL of a cell suspension containing 10^6^ viable A549 cells. The mice did not show an apparent mass on day 1. The development of tumor lesions was examined on day 7. The mice that showed a distinct tumor 4–5 mm in diameter [90] were divided into four groups. Starting on day 7, all mice that had developed apparent tumors were randomly divided into four groups (8 per group). Group I, the control group, was given saline on a daily basis through an intraperitoneal injection. Group II received intraperitoneal injections of 5-FU (100 mg/kg/week) (Sigma; St. Louis, MO, USA). Group III received intraperitoneal injections of zotarolimus (2 mg/kg/day) (MedChemExpress, Monmouth Junction, NJ, USA). Group IV was given intraperitoneal injections of zotarolimus (2 mg/kg/day) and 5-FU (100 mg/kg/week). We determined the 5-FU dosage on the basis of mouse studies on the role of 5-FU in the apoptosis, invasion, metastasis, angiogenesis, and growth signaling mechanisms of cancer cells [91,92,93]. In addition, the dosage was lower than the threshold at 150 mg/kg/week; the increased efficacy correlated with an increase in the lethal toxicity of 5-FU [94]. Alternatively, the dosage of zotarolimus was based on that generally administered for everolimus [95]. Tumor growth was monitored every seven days by measuring the largest and smallest diameters. Tumor volume was calculated using the following formula: V = 0.5 × a × b^2^, where a is the largest diameter and b is the smallest.

### 4.3. Clinical Observations and Histopathological Analysis

The mice were observed on a daily basis for clinical signs and sacrificed after one month. Body weight was measured every three days. Blood was collected to evaluate hematology under anesthesia at the end of the treatment period. Tumor specimens were divided into two groups. One group was treated with 10% formalin and embedded in paraffin. The specimens were assessed by hematoxylin and eosin staining; terminal deoxyribonucleotidyl transferase (TdT)-mediated biotin-16-dUTP nick-end labeling (TUNEL assay; APO-BrdU™TUNEL Assay Kit, BD Pharmingen Inc., San Diego, CA, USA); and immunohistochemistry (IHC) for IL-1β, TNF-α, TGF-β, and CD44. Additional IHC staining for IL-1β, TNF-α, TGF-β, and CD44 in the tumor was assessed using primary antibodies against IL-1β, TNF-α, TGF-β, and CD44 (Merck, Billerica, MA, USA). Protein expression was measured through IHC using the TAlink mouse/rabbit polymer detection system by BioTnA (Kaohsiung, Taiwan). A Moticam 2300 (Motic Instruments, Richmond, BC, Canada), a high-resolution digital microscope equipped with Motic Images Plus (version 2.0), was used to capture and analyze the images. The other group was preserved in a freezer at −80 °C. Western blotting was applied to examine cleaved caspase 3, ERK, Bcl-2, COX-2, VEGF, and EGFR.

### 4.4. Measurement of Serum Levels of IL-6 and IL-10

The blood samples were used to measure serum levels of IL-6 and IL-10 using mice commercial kits (ab213749 and ab100697; Abcam, Cambridge, MA, USA) following the manufacturer instructions. Samples of 50 μL were added to each well of the 96-well antibody-coated plates and then incubated for 2 h at room temperature. The detector antibody solution (50 μL) was loaded into each well and the plates were incubated for 1 h at room temperature. Next, 50 μL of the HRP-Streptavidin solution (ab210901, Abcam) was added, and the plates were incubated for another 1 h. Then, 100 μL of TMB substrate was added to each well and the plates were incubated for 10 min in the dark. The reaction was stopped with the addition of 100 μL stop solution. A wavelength 450 nm was used to read the absorbance and the results are expressed in pg/mL.

### 4.5. RNA Extraction and Real-Time Quantitative PCR

TRI Reagent (Sigma-Aldrich) was used to extract total RNA from the tumor tissues. We assessed RNA concentration on the basis of absorbance at 260–280 nm and 230–260 nm on a Qubit fluorometer (Invitrogen, Carlsbad, CA, USA). The RNA (1 μg) was reverse transcribed into cDNA using an iScript cDNA synthesis kit (Bio-Rad, Hercules, CA, USA) following the manufacturer’s protocol. Real-time PCR was performed using the cDNA and iTaq universal SYBR Green supermix (Bio-Rad) following the manufacturer’s instructions. NF-κB and IκBα mRNA expression levels were determined using the CFX Connect Real-Time PCR Detection System (Bio-Rad). The settings for the PCR were as follows: 40 cycles of 95 °C for 30 s; 95 °C for 15 s, and 60 °C for 30 s; and a final 5 min at 72 °C. The NF-κB, IκBα, and *β-actin* sequence primers employed in this study were NF-κB forward, 5′-ATGGCTTCTATGAGGCTGAG-3′, and reverse, 5′-GTTGTTGTTGGTCTGGATGC-3′; *IκB-α* forward, 5′-GCCCTTGTCCCTGTCCCTA-3′, and reverse, 5′-GCAGAGTATTTCCCTTTGGTTTGA-3′; and *β actin* forward, 5′-ACTGGAACGGTGAAGGTGACA-3′, and reverse, 5′-ATGGCAAGGGACTTCCTGTAAC-3′ [96,97]. The expression levels for each target gene were calculated relative to the *β-actin* levels and expressed using the 2^−ΔΔCt^ method.

### 4.6. Western Blotting

Following the experiment, mice were euthanized with an overdose of anesthetic combined with carbon dioxide. Tumors were quickly obtained from the mice, then coarsely minced and homogenized. Western blotting was then performed as previously described [98,99,100]. For detection, we used antibodies against β-actin, Bcl-2, and COX-2 from Sigma-Aldrich Inc. (St. Louis, MO, USA); VEGF and EGFR were also purchased from Sigma-Aldrich (St. Louis, MO, USA); and cleaved caspase 3, phosphorylated ERK (threonine 202/tyrosine 204), and ERK were from Cell Signaling Technology (Beverly, MA, USA). Enhanced chemiluminescence reagents (Thermo Scientific, Rockford, MA, USA) generated the immunoreactive signals and UVP ChemStudio (Analytik Jena, Upland, CA, USA) was used for signal detection. The quantification of protein expression and phosphorylation was performed using ImageJ software from the National Institutes of Health (NIH; Bethesda, MA, USA).

### 4.7. Statistical Analysis

All results are shown as mean ± standard deviation. A *t*-test was used to determine if there was a difference between two groups. When more than two groups were analyzed, an ANOVA test using a post hoc Bonferroni correction determined differences between the groups. *p*-values less than 0.05, 0.01, and 0.001 were considered significant, very significant, and extremely significant, respectively.

## 5. Conclusions

In conclusion, this study highlights the potential co-effect of zotarolimus and 5-FU for the treatment of human lung adenocarcinoma. Our findings evidenced that zotarolimus alone inhibits the tumor development of A549 cell in vivo and that combining zotarolimus with a traditional chemotherapy drug such as 5-FU exerts a stronger inhibitory effect. This observation is associated with the inhibition of tumor growth via an apoptotic mechanism in A549 cells where the apoptotic expression of cleaved caspase 3 and ERK phosphorylation is enhanced and the anti-apoptotic expression of Bcl-2 is reduced. Our study also revealed that both zotarolimus alone and the combination of zotarolimus and 5-FU decreased the production of inflammatory cytokines, including IL-1β, TNF-α, and IL-6, and increased the production of the anti-inflammatory cytokine Il-10. This could inhibit tumorigenesis. Anti-tumor and anti-inflammation treatments are characterized by the decreased accumulation of COX-2 and NF-κB mRNA and an increase in IκBα mRNA. The inhibition of the mTOR signaling pathway in tumors by zotarolimus reduced TGF-β, CD44, VEGF, and EGFR expression, which could decelerate tumor cell migration and invasion in cancer metastasis. Collectively, the in vivo data revealed the pharmacological effects of zotarolimus, where it was shown effectively to enhance apoptosis, decrease inflammatory processes, and increase cancer metastasis-suppressing mechanisms. This indicates certain benefits for tumor treatment revealed by the xenograft-induced tumor mass and volume in the nude mice. The combination of zotarolimus and 5-FU has a synergistic effect on suppressing tumor growth, although the use of zotarolimus alone also has a similar effect. These results may pave the way for providing more options in cancer treatment and the development of new therapeutic strategies for treating lung adenocarcinoma cancer.

## Figures and Tables

**Figure 1 ijms-22-04562-f001:**
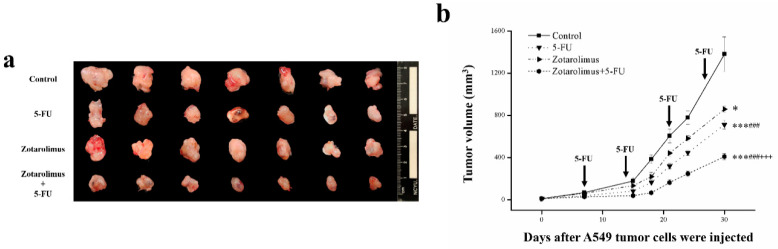
Relative volume of tumors. (**a**) Photographs of tumors excised after sacrifice on day 30. (**b**) Volumes of the A549 tumor masses from BALB/cByJNarl mice given different treatments: control (saline), 5-FU (100 mg/kg/week), zotarolimus (2 mg/kg/day), and zotarolimus (2 mg/kg/day) combined with 5-FU (100 mg/kg/week). All data are presented as mean ± standard deviation, *n* = 7 per group. * *p* < 0.05 and *** *p* < 0.001 compared with the control group. ^###^
*p* < 0.001 compared with the 5-FU-treated group. ^+++^
*p* < 0.001 compared with the zotarolimus-treated group. All treatments were initiated on day 7 when tumors were detected.

**Figure 2 ijms-22-04562-f002:**
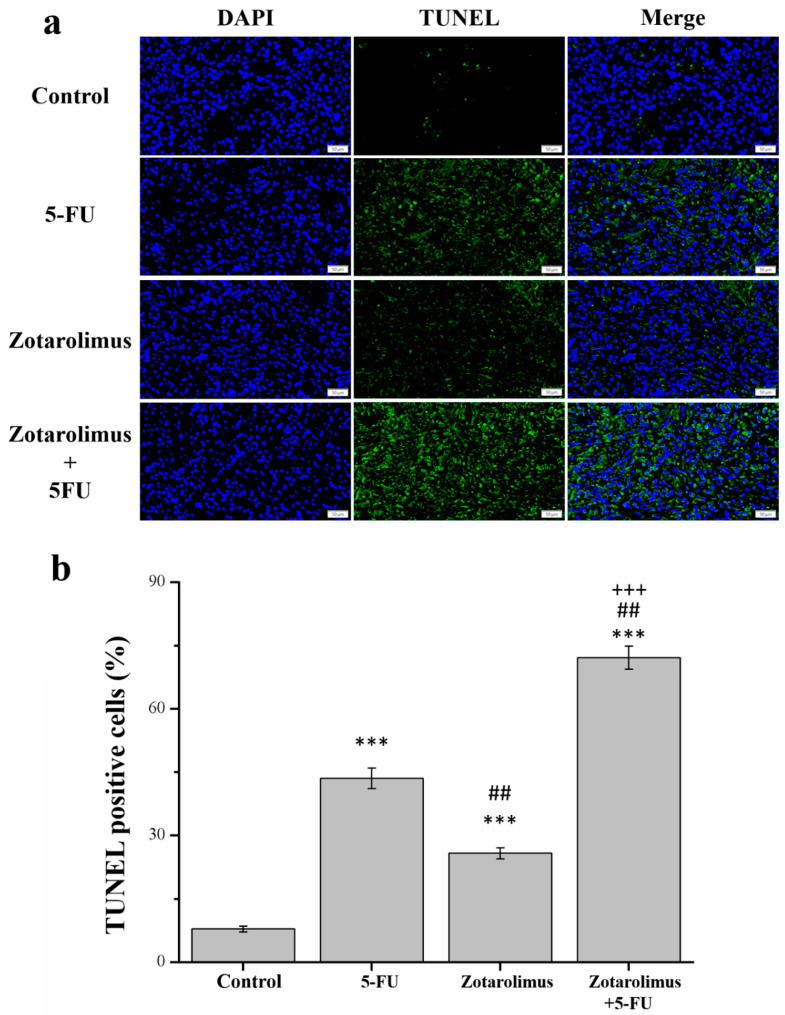
Analysis of apoptosis in A549 tumors by TUNEL/DAPI staining (×200). (**a**) Images of TUNEL and DAPI staining and merge. (**b**) The percentage of TUNEL positive cells in the A549 tumor mass of control BALB/cByJNarl mice and mice given different treatments: 5-FU (100 mg/kg/week), zotarolimus (2 mg/kg/day), and zotarolimus (2 mg/kg/day) combined with 5-FU (100 mg/kg/week). All data are presented as mean ± standard deviation, *n* = 7 per group. *** *p* < 0.001 compared with the control group. ^##^
*p* < 0.01 compared with the 5-FU-treated group. ^+++^
*p* < 0.001 compared with the zotarolimus-treated group. All treatments were started on day 7 when tumors were detected. Scale bars = 50 µm.

**Figure 3 ijms-22-04562-f003:**
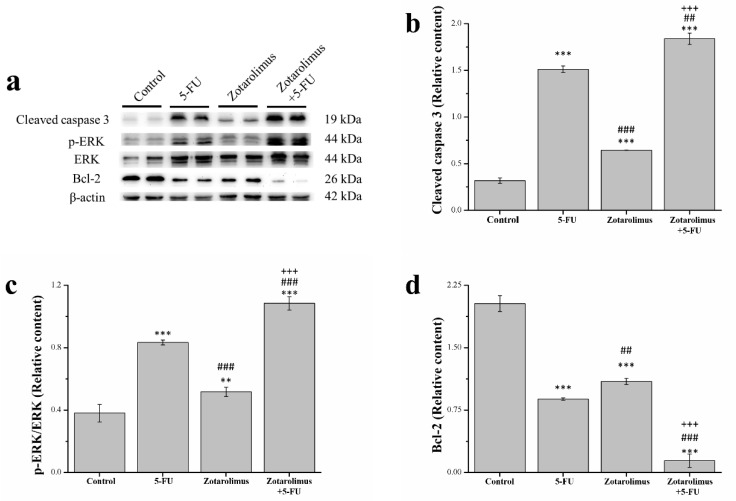
Analysis of apoptosis-related proteins including (**a**) a representative western blot showing the levels of apoptosis-related proteins extracted from the tumors; (**b**) cleaved caspase 3 expression; (**c**) ERK phosphorylation; and (**d**) Bcl-2 expression in the A549 tumor masses taken from BALB/cByJNarl control mice and mice given different treatments: 5-FU (100 mg/kg/week), zotarolimus (2 mg/kg/day), and zotarolimus (2 mg/kg/day) combined with 5-FU (100 mg/kg/week). All data are presented as mean ± standard deviation, *n* = 7 per group. ** *p* < 0.01 and *** *p* < 0.001 compared with the control group. ^##^
*p* < 0.01 and ^###^
*p* < 0.001 compared with the 5-FU-treated group. ^+++^
*p* < 0.001 compared with the zotarolimus-treated group.

**Figure 4 ijms-22-04562-f004:**
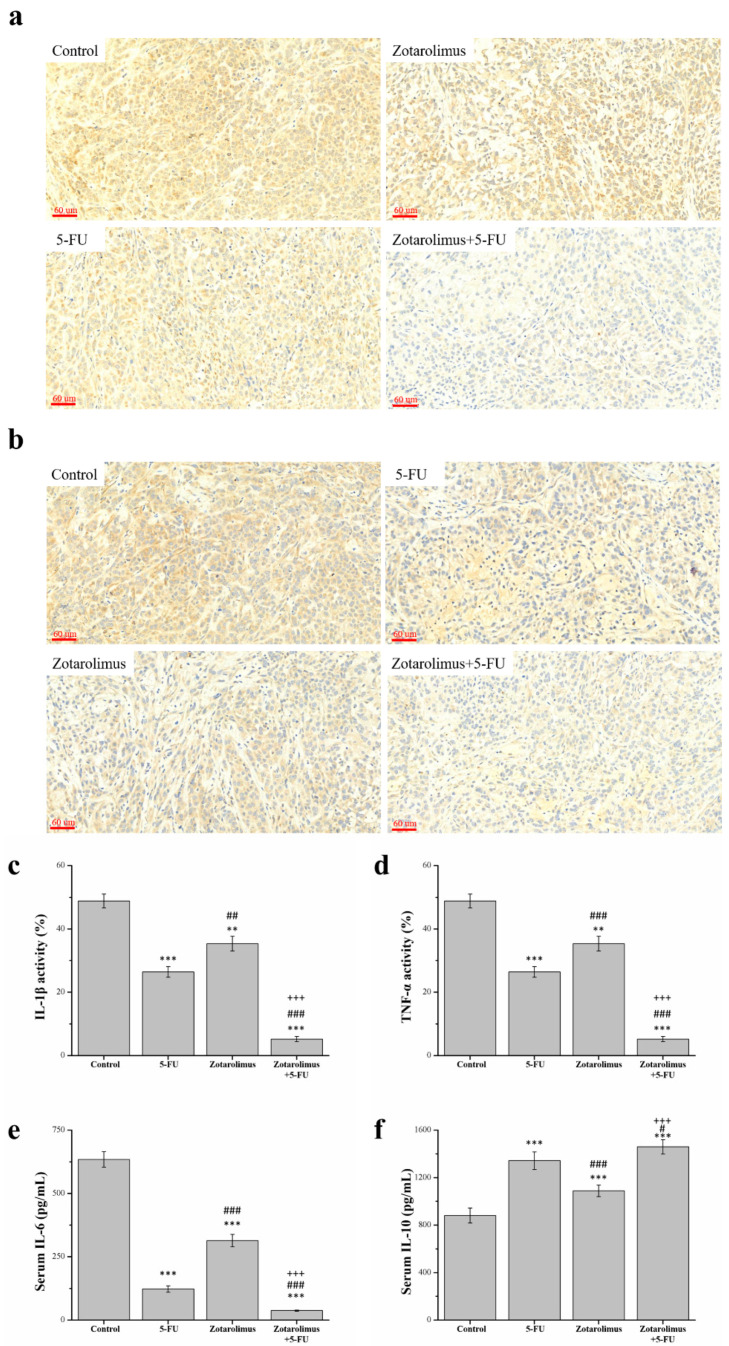
Immunohistochemical expression of (**a**) IL-1β and (**b**) TNF-α in A549 tumors from BALB/cByJNarl mice and the comparative immunohistochemical expressions of (**c**) IL-1β and (**d**) TNF-α; as well as (**e**) IL-6 and (f) IL-10 levels in serum. All experiments were conducted on the A549 tumor masses from BALB/cByJNarl mice. Findings are reported for the control group and treatments: 5-FU (100 mg/kg/week), zotarolimus (2 mg/kg/day), and zotarolimus (2 mg/kg/day) combined with 5-FU (100 mg/kg/week). All data are presented as mean ± standard deviation, *n* = 7 per group. ** *p* < 0.01 and *** *p* < 0.001 compared with the control group. ^#^
*p* < 0.05, ^##^
*p* < 0.01, and ^###^
*p* < 0.001 compared with the 5-FU group. ^+++^
*p* < 0.001 compared with the zotarolimus group. Scale bars = 60 µm.

**Figure 5 ijms-22-04562-f005:**
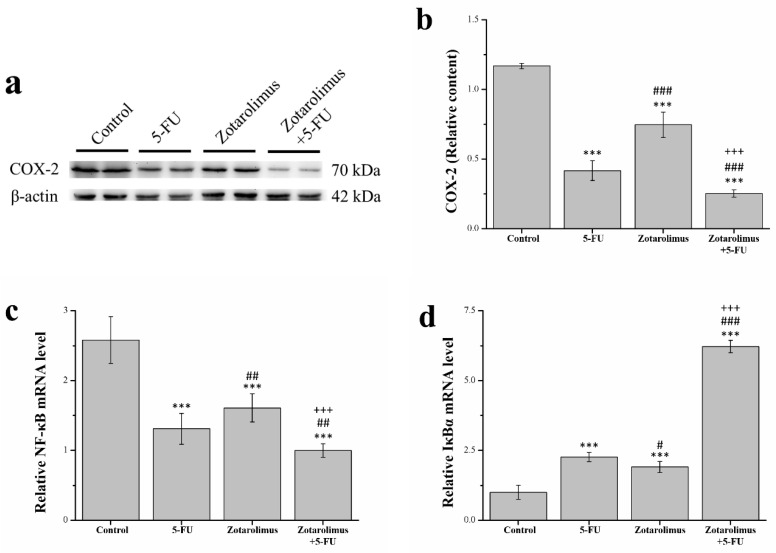
Western blot analysis of (**a**) inflammation-related factors extracted from the tumors in a representative blot; and (**b**) COX-2 expression in the A549 tumor mass from BALB/cByJNarl mice subjected to various treatments. The relative mRNA expression levels of (**c**) NF-κB and (**d**) IκBα were estimated using quantitative PCR. All experiments were conducted on A549 tumor masses from BALB/cByJNarl mice in the control group and mice given different treatments: 5-FU (100 mg/kg/week), zotarolimus (2 mg/kg/day), and zotarolimus (2 mg/kg/day) combined with 5-FU (100 mg/kg/week). All data are presented as mean ± standard deviation, *n* = 7 per group. *** *p* < 0.001 compared with the control group. ^#^
*p* < 0.05, ^##^
*p* < 0.01, and ^###^
*p* < 0.001 compared with the 5-FU group. ^+++^
*p* < 0.001 compared with the zotarolimus group.

**Figure 6 ijms-22-04562-f006:**
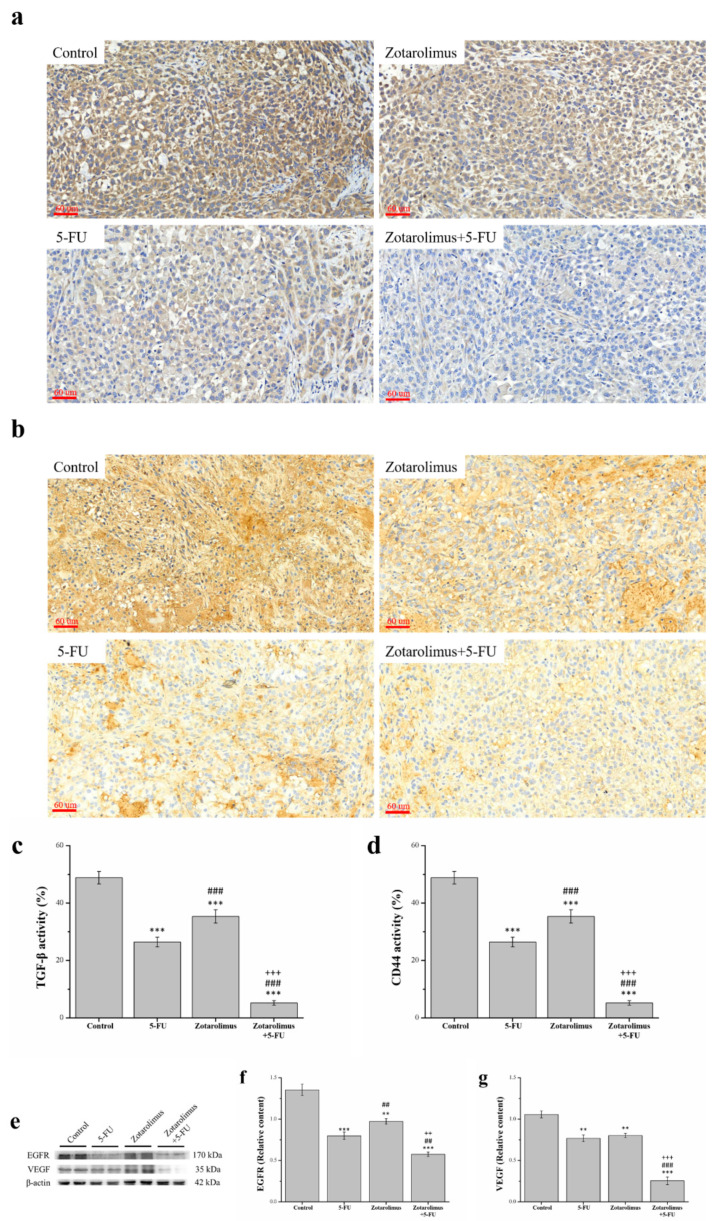
Immunohistochemical expression of (**a**) TGF-β and (**b**) CD44 and a comparison of the expression levels of (**c**) TGF-β and (**d**) CD44. Western blot analysis (**e**) showing the expression of metastasis-related factors extracted from tumors in a representative blot. Expression of (**f**) VEGF and (**g**) EGFR in the A549 tumor masses from BALB/cByJNarl in control mice and mice given different treatments, including 5-FU (100 mg/kg/week), zotarolimus (2 mg/kg/day), and zotarolimus (2 mg/kg/day) combined with 5-FU (100 mg/kg/week). All data are presented as mean ± standard deviation, *n* = 7 per group. ** *p* < 0.01 and *** *p* < 0.001 compared with the control group. ^##^
*p* < 0.01 and ^###^
*p* < 0.001 compared with the 5-FU-treated group. ^++^
*p* < 0.01 and ^+++^
*p* < 0.001 compared with the zotarolimus-treated group. Scale bars = 60 µm.

**Table 1 ijms-22-04562-t001:** Weight of the tumors from BALB/cByJNarl mice after sacrifice.

	Tumor Weight (g)	Tumor Inhibition Rate (%)
Control	1.248 ± 0.082	-
5-FU	0.687 ± 0.033 ***	50.0
Zotarolimus	0.859 ± 0.035 *** ^##^	31.2
Zotarolimus + 5-FU	0.416 ± 0.026 *** ^###^ ^+++^	66.7

All data are presented as mean ± standard deviation, *n* = 7. *** *p* < 0.001 compared with control group. ^##^
*p* < 0.01 and ^###^
*p* < 0.001 compared with 5-FU-treated group. ^+++^
*p* < 0.001 compared with the zotarolimus-treated group.

## Data Availability

The data presented in this study are available on request from the corresponding author.

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
