# Peer review of "The Anti-Cancer Effects of a Zotarolimus and 5-Fluorouracil Combination Treatment on A549 Cell-Derived Tumors in BALB/c Nude Mice"

_ijms, 2021, doi:10.3390/ijms22094562_

Round 1

Reviewer 1 Report

The authors focused on the anti-cancer effects of zotarolimus in A549, especially apoptosis, cytokine expression and metastasis-associating factors expression. I think the concept of this project is interesting. However, the clinical insight and novelty should be clarified more clearly and the conclusions seem not to be fully supported by the presented data. There are some major concerns listed as follows.

Major point

Introduction part

1) You should mention the reason why you focused on not only zotarolimus alone but also the combination with 5-FU.

2) Page 3, line 100. The reference 24 reported the utility of everolimus, not zotarolimus. Therefore, the sentence “zotarolimus can be potentially …….of lung cancer” is misleading.

 I expected that you focused on zotarolimus because there are previous reports about the utility of everolimus as anti-cancer agent. However, to the best of my knowledge, there have been no reports regarding the anti-cancer effect of zotarolimus in malignant neoplasms. You should added this point as a novelty of this study, and in addition, describe the difference and/or overlap mechanism between zotarolimus and everolimus in detail.

Results

3) Basically, you should insert the specific p-value in the main text. e.g) Page 4, line 138: “Our findings indicated that ………combined with 5-FU (p=xx, xx, and xx), respectively (Figure 2a)”

4) In several sentences, what you compared with is unclear. e.g.) Page 4, line 163:  …a lower expression of caspase 8 and ERK phosphorylation than what?

5) The dose of 5-FU (100mg/kg) is higher than usual. I know that this is for experiment, but you should add this point as a limitation, and mention whether side effects were observed in mice or not.

6)Page 4, line 143: I cannot understand the meaning of this sentence “Moreover, the number s of TUNEL positive cells ………higher than those in mice treated with 5-FU and zotarolimus”. It seems to be inconsistent with the figure 2b for me.

7) You investigated the expression change of many factors, however, it was not examined whether these changes really induced the change of immune microenvironment and/or metastatic status.

 Section 2.5: Please examine the histological changes of immune microenvironment such as TIL infiltration levels after the treatment.

 Section 2.6: Please examine whether the metastasis was inhibited after the treatment or not.

 As long as you examined these points, you cannot say that “zotarolimus combined with 5-FU exhibited excellent inhibition of A549 cell growth in terms of apoptosis, inflammation, and metastasis” (Page 12, line 460).

8) Figure 3. Statistical analyses showed significant changes of bcl-2 between control and zotalorimus alone groups, but it seems not to be significant in westernblotting (Figure 3a). In addition, caspase 8 did not show significant changes between these two groups. I wonder if zotalorimus really induced apoptosis or not. Are there a possibility that zotalorimus interact with 5-FU? Please discuss this point in discussion part.

Discussion

9) You discussed the mechanism of the anti-cancer effects of zotarolimus by focusing on the interaction of examined factors and other many factors in detail. However, most of these ideas were not supported by the results of this present study. (Actually, you should perform more experiments using siRNA and so on to prove these ideas.) Please add this point as limitation, and make the discussion part shorter than current version.

Materials and methods

10) In each section, you selected several factors among cytokines and metastasis-associating factors. Please clarify the reason why you chose them. e.g.) Why didn’t you choose well-known EMT markers (such as ZEB1 and others) for the examination regarding metastasis?

Minor point

11) Figure 2b: The base is not zero. Is it correct?

Reviewer 2 Report

The paper by Chang et al with the title “The anti-cancer effects of a zotarolimus and 5-fluorouracil com-2 bination treatment on A549 cell-derived tumors in BALB/c 3 nude mice” address targeting mTOR in combination with fluorouracil in lung cancer.

The author shows an effect of single and combination treatments on tumor progression. The treatment increases cell death in the tumor, decreases inflammation and down regulation of pathways related to cell migration. Overall, the data are sound and suitable for publication. I still find that the paper could be improved and I have listed different suggestions.

Major points:

  1. I miss a control in the study, where the mice have been treated with rapamycin. The authors claim that Zotarolimus is a better drug due to shorter half-life. This is unclear to me, and the paper will benefit with clarification on that point in the introduction and discussion.

  1. The treatment shows increased p-ERK, which the authors claim to be a marker for cell death. P-ERK can be stress induced but can also be related to cell proliferation. Staining for Ki67 could be informative and complement the PCNA blot.

  1. The authors use immune deficient mice but have not addressed, which parts of the immune system are compromised. I think this should be addressed in the discussion and in context to their findings. Furthermore, it should be pointed out if they measure human or mouse cytokines. This is also important for the qPCR data – Are we looking at human or mouse gene expression?

  1. The number of samples that have been quantified for the WB is not clear for me. Is it n=2 or n=7? If it is 7 samples, the un-cut gels should be presented.

  1. The IHC stainings are hard to see and interpret. Please use high magnification and increased resolution.

  1. A WB for cleaved caspase 3 will complement the main findings, where cell death is induced in the tumors.

  1. The mice have been treated for a long time with one or two drugs. Did the mice weight decrease and was the welfare of the mice compromised? This are important issues, as the authors are doing a pre-clinical study.  

Minor points:

  1. The use of 5-FU is not introduced in the introduction but first presented in the discussion. I think this part should be moved to the introduction:

5-FU is a pyrimidine analog widely used to treat cancers. 454 However, drug resistance is a critical factor limiting the clinical use of 5-FU to treat lung 455 cancer [71]. Therefore, this study examined the anti-tumor effects of the mTOR inhibitor, 456 zotarolimus, on the A549 cell, and the synergistic effects of zotarolimus combined with 5-457 FU.

  1. Each paragraph in the result section could benefit with a one sentence introduction. Hereby the reader can follow the rationality of the experiment.

Round 2

Reviewer 2 Report

The authors have address the majority of the questions. I find that the manuscript has improved and they have added new data that are improving the study.

However, part of point 3 has not been address:
3. Furthermore, it should be pointed out if they measure human or mouse
cytokines. This is also important for the qPCR data – Are we looking at human or mouse gene
expression?
